Personality predicts the propensity for social learning in a wild primate

Carter Alecia J. 1 2 3 ac854@cam.ac.uk
Marshall Harry H. 2 4
Heinsohn Robert 1
Cowlishaw Guy 2
1 The Fenner School of Environment and Society, The Australian National University , Acton, Canberra, ACT , Australia
2 The Institute of Zoology, Zoological Society of London , Regent’s Park, London , UK
3 Large Animal Research Group, Department of Zoology, University of Cambridge , Cambridge , UK
4 Division of Ecology and Evolution, Department of Life Sciences, Imperial College London , Silwood Park, Berkshire , UK
Garant Dany
Electronic publication date: 2014 Mar 11
Publication date: 2014
Volume: 2
Electronic Location ID: e283
Received 2013 Nov 11; Accepted 2014 Jan 30
Copyright: © 2014 Carter et al.
Copyright year: 2014
Copyright holder: Carter et al.
License: This is an open access article distributed under the terms of the Creative Commons Attribution License, which permits unrestricted use, distribution, and reproduction in any medium, provided the original author and source are credited.
License URL: https://creativecommons.org/licenses/by/3.0/

Keywords: Baboon, Information use, Papio ursinus, Personality, Social information, Social learning

Funding: The Leakey Foundation, the Animal Behavior Society (USA), the International Primatological Society, and the Explorers Club Exploration Fund Fenner School of Environment and Society studentship NERC Open CASE Financial support was provided by grants from the Leakey Foundation, the Animal Behavior Society (USA), the International Primatological Society, and the Explorers Club Exploration Fund to AJC. AJC was supported by a Fenner School of Environment and Society studentship. HHM was supported by a NERC Open CASE studentship. The funders had no role in study design, data collection and analysis, decision to publish, or preparation of the manuscript.

==============================
Social learning can play a critical role in the reproduction and survival of social animals. Individual differences in the propensity for social learning are therefore likely to have important fitness consequences. We asked whether personality might underpin such individual variation in a wild population of chacma baboons (Papio ursinus). We used two field experiments in which individuals had the opportunity to learn how to solve a task from an experienced conspecific demonstrator: exploitation of a novel food and a hidden item of known food. We investigated whether the (1) time spent watching a demonstrator and (2) changes in task-solving behaviour after watching a demonstrator were related to personality. We found that both boldness and anxiety influenced individual performance in social learning. Specifically, bolder and more anxious animals were more likely to show a greater improvement in task solving after watching a demonstrator. In addition, there was also evidence that the acquisition of social information was not always correlated with its use. These findings present new insights into the costs and benefits of different personality types, and have important implications for the evolution of social learning.

Introduction

Individuals can acquire information either through interacting with their environment directly—personal information—or through observing the behaviour of other individuals—social information (Dall et al., 2005). Personal information can be costly and time consuming to collect (Laland, 2004). For social animals, social information provides an alternative to gathering costly personal information, potentially facilitating the acquisition of information about when, where and how to get food, where to travel and remain, whom to choose as a mate, and which predators to avoid (Giraldeau, Valone & Templeton, 2002; Reader & Biro, 2010). Social learning—where an individual changes its behaviour after observing another—allows the rapid dissemination of novel information among group members, and has also been widely implicated in the formation of tradition and cultures within species (Whiten, 2000; Castro & Toro, 2004). On the other hand, social learning can be conceived as a form of ‘information parasitism’ and can be maladaptive if information rapidly becomes outdated (Laland, 2004). Nonetheless individual differences in innovation and social learning abilities remain largely overlooked (Thornton & Lukas, 2012). This is surprising, as an understanding of individual variation in social learning is likely to be necessary for predicting the formation and persistence of traditions and culture within groups, and the individual fitness advantages of personal versus social information use (Thornton & Lukas, 2012).

Animal personality refers to between-individual differences in behaviour that persist through time (Sih, Bell & Johnson, 2004). Evidence for personality-related individual differences in information use (Kurvers et al., 2010; Carter et al., 2013) suggests personality may also explain individual differences in social learning (Marchetti & Drent, 2000). This may occur at two different steps in the social learning process, when (1) a subset of individuals initially gathers personal information about a novel situation and (2) this subset—the ‘demonstrators’—provide other individuals with the opportunity to then gather and use social information about the novel situation from them. In the first step, for individuals to exploit a novel food or solve a novel task they must interact with it for long enough to obtain sufficient personal information about it (Seferta et al., 2001; Thornton & Samson, 2012). Individual variation in responses to novel stimuli are frequently used to differentiate individuals by two personality traits, namely boldness and exploration (Carter et al., 2012b). Those animals that are less bold, i.e., shyer, or less explorative of novelty, i.e., ‘neophobic’, individuals (Réale et al., 2007; Carter et al., 2012c) tend to be less willing to interact with a novel food or task and are thus less able to acquire the personal information necessary to solve the problem (but see Cole & Quinn, 2011). For example, neophobic myna (Acridotheres tristis) are less likely to solve innovation tasks than neophilic myna (see also Coleman, Tully & McMillan, 2005; Toxopeus et al., 2005; Sol et al., 2011). In the second step, those animals that are less willing or able to gather personal information may be more likely to use social information. For instance, where individuals seek to exploit a novel food or solve a novel task, bolder individuals may do so directly because they are willing to gain personal information about it, and subsequently become demonstrators for shyer individuals who may be more likely to attend to, and act upon, social information (Kurvers et al., 2010).

We investigated the relationship between animal personality and social learning in wild chacma baboons (Papio ursinus). Baboons present an ideal system to explore such individual differences, since they are highly social, live in stable groups, and social learning has previously been documented among infants and juveniles (Whiten, 2000). In addition, individual baboons in our study population vary consistently in two unrelated personality traits, boldness and anxiety, that are repeatable over a three-year period and for which all individuals in the study groups have been assayed up to six times (Carter et al., 2012c). We use the term ‘boldness’ to refer to an individual’s response to a novel food, where bolder individuals spend more time inspecting a novel food; and ‘anxiety’ for an individual’s response to a threatening stimulus, where more anxious individuals show a stronger fear response towards, and spend more time investigating, a threat (Carter et al., 2012c). Though threat responses can also be used to describe boldness, we have previously found no evidence that our threat presentation test measures boldness in this species (for further details, see Carter et al., 2012c).

To test our hypothesis that personality traits (boldness, anxiety) influence individual propensity for social learning, we performed two types of field experiments in which social learning to solve a task could occur: exploitation of a novel food (experiment NF) and a hidden food (HF). We investigated whether (Q1) propensity to solve the tasks using personal information alone, (Q2) time spent collecting social information (watching a demonstrator solve the tasks), and (Q3) social learning, i.e., changes in task-solving behaviour after exposure to a demonstrator, were related to the baboons’ boldness or anxiety. Thus we addressed three questions with two experiments for two personality traits that could influence social learning.

We had several predictions in line with our questions. First, we predicted that bolder baboons would be more likely to ‘solve’ the hidden food task as bolder individuals are more willing to interact with a novel food (Carter et al., 2012a) (prediction 1 for boldness: P1HFb). We further predicted that there would be no relationship between anxiety and the propensity to solve the hidden food task as anxiety is unrelated to boldness in this species (Carter et al., 2012c) and anxious individuals are otherwise no more likely to interact with a novel food than calm individuals (P1HFa). Second, regarding whether an individual would collect social information, we predicted that more anxious baboons would spend more time watching the demonstrator than calmer baboons, in both the novel food and hidden food tasks, as anxious baboons attend to a threatening stimulus for longer than calm baboons (Carter et al., 2012c) (P2NF,HFa). We further predicted that shyer baboons would spend more time watching a demonstrator than bolder baboons in both tasks as shy geese attend to social information more than bold geese (Kurvers et al., 2010) (P2NF,HFb). Finally, regarding whether individuals differed in their propensity for social learning in the novel food and hidden food tasks, we predicted that more anxious individuals (P3NF,HFa) would show greater social learning than calmer baboons, as anxious individuals were predicted to attend to social information more than calm baboons (P2NF,HFa above). Similarly, we predicted that shyer individuals (P3NF,HFb) would show greater social learning than bold baboons in both tasks, as shy individuals were predicted to attend to social information more than bold baboons (P2NF,HFb above).

Materials and Method

Study area and species

We studied the chacma baboons over three years, from May to November 2009, May to October 2010 and June to September 2011 at Tsaobis Leopard Park, Namibia (15°45′E, 22°23′S). Two groups of baboons (n = 44, 31 in 2009) have been habituated to the presence of observers at close range and are individually recognisable. We collected data annually from 57 adult, subadult and juvenile baboons (we did not test individuals who were dependent young in 2009) over the 2009 to 2011 period. Individuals were assigned to age classes (juvenile/adult) on the basis of canine eruption in males and menarche in females (after which they were considered adult). Age (years) was estimated from a combination of known birth dates and dental patterns of tooth eruption and wear (Huchard et al., 2009). Individual ranks were assessed through dominance interactions that were recorded during focal observations and ad libitum using Matman 1.1.4 (Noldus Information Technology, Wageningen, The Netherlands). Hierarchies in both troops were strongly linear in all years (in 2010, for example, Landau’s corrected linearity index: hlarger troop′=0.71, hsmaller troop′=0.82, p < 0.001 in both). Rank was expressed as a relative rank (which controls for group size), calculated from absolute ranks for each baboon using the formula 1−[(1−r)/(1−n)] where r is the individual’s absolute rank and n is the group size of the individual.

Personality assessments

Boldness and anxiety were assessed using an experimental approach in 2009, 2010 and 2011 (for further details, see Carter et al., 2012a; Carter et al., 2012c). Boldness was assessed by scoring responses to a novel food, while anxiety was assessed by scoring responses to a venomous snake (Carter et al., 2012c). In all cases, individuals were presented with a stimulus when they were alone and moving between food patches. The stimuli were presented on the edges of game trails and paths regularly used by the baboons. All experiments were filmed to facilitate data extraction (Panasonic SDR-SW20, Kadoma, Osaka, Japan; see movie files in Carter et al., 2012c).

Stimuli for the boldness experiments consisted of novel food items which included hard-boiled eggs with the shell on or removed, or a small egg-shaped bread roll, all of which were dyed red or green (Moir’s food dye), in 2009; semi-dried eighths of apple or pear, dyed red, in 2010; and eighths of an orange or equivalent-sized pieces of butternut squash in 2011. Any naïve individual that saw another individual interacting with a novel food was presented with a different novel food when they were tested. We recorded the latency in seconds (s) to approach the food item on detection (if the food item was not approached, the individual was given the maximum value of 150 s; individuals that did not detect the item were subsequently re-tested), the time spent inspecting the food item (s; the time between approaching the food item and the end of the experiment, either leaving or eating the item), and the time spent handling the food item (s; the time spent touching the food item). Fifty-eight, fifty-four and fifty baboons received novel food presentations in 2009, 2010 and 2011, respectively (mean and median number of presentations = 2.8, 3.0; range = 1–3 presentations).

The stimulus for the anxiety experiments was a taxidermic puff adder Bitis arietans. We recorded as binary responses whether an individual approached, vocalised, performed a self-directed behaviour, bared its teeth, flagged its tail, stopped and backed away (see Carter et al., 2012c for details). If more than one baboon saw the stimulus during a given presentation, we extracted data for both of the individuals if the second individual did not respond to the reaction of the first individual (and in one case third individual). A baboon was recorded as responding to another individual if it looked in the direction of and/or approached the individual who initially saw the stimulus. In total, we completed 153 snake presentations to 57 baboons over 2009 to 2011 (mean and median number of presentations = 2.7, 3.0; range = 1–6 presentations).

As both boldness and anxiety have been shown to be repeatable across years (r = 0.26, 0.34, respectively) (Carter et al., 2012c), we used a composite score for each trait from trait values collected over three years (see Carter et al., 2012c). Boldness and anxiety are independent in this species, thus a bolder baboon is equally likely to be as anxious as a shyer baboon (Carter et al., 2012c).

Social learning experiments

From 2009 to 2011, we performed two types of experiment designed to assess differences in social learning in baboons: a novel food experiment and a hidden food experiment. In both cases, we attempted to adopt a ‘test–learn (via a demonstrator)–re-test’ experimental design. The experiments were run concurrently to limit order effects. All experiments were filmed to facilitate data extraction, and the baboons’ identities and activities were dictated to the camera (Panasonic SDR-SW20, Kadoma, Osaka, Japan; see Video S1). The study population in 2010 comprised 21 male and 34 female baboons who were on average (mean) 8.2 years old (range: 4–21), had a relative rank of 0.5 (0–1), inspected the novel food on average for 20 s (0–120) with an mean boldness score of 0.46 (−3.90–1.47) (note that low values represent individuals that inspected the food for longer, such that high scores indicate shyer animals) and an average anxiety score of −0.02 (−0.22–0.53) (where high scores indicate more anxious animals).

Novel food experiments

The ‘test’ stage of the novel food experiments was also our assay for boldness (see above). Thus, solitary individuals were presented with a novel food while moving between natural food patches. In the majority of cases, the test subjects only briefly inspected the food and failed to eat it (Carter et al., 2012a). In the ‘learn’ stage, the same individuals were subsequently given the opportunity to watch an experienced demonstrator with the same novel food. The demonstrator was an individual who had encountered the novel food at least once before, and had consumed at least part of it on the last encounter. We achieved this by presenting a novel food item to small subgroups (two-five baboons within 5 m of one another) that included a randomly-chosen demonstrator. We recorded the identities of any additional baboons that came within 5 m of the demonstrator (and could therefore watch the demonstrator) and the duration that the (potential) observers watched the demonstrator (seconds). Watching the demonstrator was defined as an individual directing its gaze towards the demonstrator. The difference in dominance ranks between the observers and the demonstrator was also recorded.

In the final ‘re-test’ stage, we subsequently presented the non-demonstrators with the novel food again, under the same conditions as in the initial test stage (i.e., while each was solitary and moving between food patches). All observers were tested after their ‘learn’ trial, but could also have been observers in others’ ‘learn’ trials subsequently (as we could not control subgroup formation). If the non-demonstrators had learned from watching the demonstrator, we expected their responses to the novel food to differ from that previously observed. Specifically, we expected shy and anxious individuals’ handling time, which positively correlates with the probability of eating the novel food (Carter et al., 2012a), to increase on the ‘re-test’ trial compared to the initial trial, while bold and calm individuals’ behaviour would not change (Table 1).

Table 1 Factors affecting task solving propensity (Q1), the attention given to a demonstrator (Q2), and social learning (Q3) in wild chacma baboons for two experiments (novel food).

Significance of fixed terms was determined by t- and p-values for GLMMs (<0.05) and z-values for LMEs (>2.00), and random terms by a log-likelihood ratio test. The reference categories are: adult (for Age class), female (for Sex), test (for Test) and control (for Condition).

Experiment	Question	Response	Nobs, Nind
(Ndemonstrators)	Deviance	Term	β	S.E/S.D.	Test
statistic	p	
Novel food	2	Time spent watching
demonstrator (s)	54, 36, (14)	180.5	Intercept	2.59	0.52	5.02		
					Anxiety	−2.79	0.51	−2.01		
					Age class: juvenile	1.75	0.40	4.36		
					Rank	−1.74	0.72	−2.43		
					Random: baboon	0.00	0.00	0.00	1.00	
					Random: troop	1.68	0.51	0.75	0.39	
					Random: demonstrator	0.00	0.00	0.06	0.81	
	3 (boldness)	Time handling food (s)	90, 32	1206.00	Intercept	1.00	0.47	2.11	<0.03	
					Test: re-test	−0.78	0.08	−9.57	<0.001	
					Boldness	−1.91	0.51	−3.77	<0.001	
					Condition: treatment	0.17	0.16	−1.09	0.28	
					Test: re-test *
Boldness	−0.14	0.05	−2.69	0.007	
					Test: re-test *
Condition: treatment	0.84	0.10	8.58	<0.001	
					Boldness *
Condition: treatment	−0.20	0.25	−0.79	0.43	
					Test: re-test *
Boldness *
Condition: treatment	−0.68	0.08	−8.67	<0.001	
					Random: baboon	5.99	2.45	3090.90	<0.001	
					Random: troop	0.00	0.00	0.00	1.00	
	3 (anxiety)	Time handling food (s)	90, 32	1301.00	Intercept	1.00	0.87	1.15	0.25	
					Test: re-test	−0.88	0.09	−9.98	<0.001	
					Anxiety	16.11	5.89	2.73	0.006	
					Condition: treatment	−0.03	0.08	−0.31	0.75	
					Age class: juvenile	3.61	1.05	3.43	<0.001	
					Test: re-test * Anxiety	−11.95	3.02	−3.96	<0.001	
					Test: re-test *
Condition: treatment	1.75	0.10	16.70	<0.001	
					Anxiety *
Condition: treatment	−8.03	3.73	−2.15	0.03	
					Test: re-test *
Anxiety *
Condition: treatment	7.66	3.04	2.52	0.01	
					Random: baboon	7.49	0.87	3430.00	<0.001	
					Random: troop	0.34	0.58	0.10	0.75	
Hidden food	1	Probability of eating
food (0/1)	80, 33	64.09	Intercept	−2.40	1.38	−0.74	0.08	
					Age class: juvenile	2.57	1.02	2.52	0.01	
					Boldness	−0.85	0.32	−2.63	0.008	
					Random: baboon	0.00	0.00	0.00	1.00	
					Random: troop	2.03	1.43	3.85	0.05	
	2	Watch (0/1)	163, 41, (18)	180.5	Intercept	−0.69	0.37	−1.85	0.06	
					Presentation number	0.21	0.08	2.82	0.005	
					Rank difference	−2.01	0.68	−2.97	0.003	
					Random: baboon	1.04	1.02	5.50	0.02	
					Random: troop	0.00	0.00	0.00	1.00	
					Random: demonstrator	0.06	0.24	0.06	0.80	
	3	Response (levels 0–4)	80, 33	208.40	Intercept	1.85	0.69	2.70		
					Boldness	−0.29	0.10	−2.81		
					Age class: juvenile	1.34	0.22	6.04		
					Random: baboon	0.06	0.25	7.29	0.007	
					Random: troop	0.85	0.92	0.00	1.00	

As a control, we also re-tested a subset of baboons that did not have the opportunity to watch a demonstrator and therefore could only change their responses to the ‘novel’ food through further personal experience. The ‘social learning of novel food’ experiment was carried out on 16 subjects in 2010; the control trials were carried out on 23 subjects between 2009 and 2011. The demonstrators for this experiment comprised 6 male and 7 female baboons who were on average 6.8 years old, had a relative rank of 0.58, inspected the novel food on the first presentation on average for 51 s with an average boldness of −1.17 and an average anxiety of 0.05. The focal subjects for this experiment comprised 13 male and 19 female baboons who were on average 7.6 years old, had a relative rank of 0.48, inspected the novel food on the first presentation on average for 19 s with an average boldness of 0.15 and an average anxiety of −0.01.

Hidden food experiments

In the hidden food experiment, the subjects were presented with a hidden-food task whilst they were moving with other baboons between food patches. The hidden-food task consisted of an empty, opaque 1 L UHT milk tetrapack that had been rinsed thoroughly and dried. One corner of the top of the tetrapack had been cut off to let the milk out, and the other corner unfolded so that the top, short end of the tetrapack was no longer flat. Fifteen to twenty dried maize kernels—a highly preferred food for the baboons—were placed in the tetrapack. Since the tetrapack was opaque, the baboons had to interact closely with it to find the food inside. We doubt that the baboons could smell the maize kernels, but if they could they would have had to interact intimately with the stimulus to do so. Individuals could discover the food by tipping the tetrapack upside down, or ripping the tetrapack open.

During the presentations we recorded the following information: the identities of the baboons that came within 5 m of the stimulus when there was no ‘demonstrator’, and thus had an opportunity to obtain personal information about the stimulus; the identities of the baboons that interacted with the stimulus, defined as approaching to smell, touch or handle the tetrapack, and whether they solved the task, defined as eating any food from the tetrapack; the identities of the baboons that came within 5 m of a demonstrator, i.e., an individual interacting with the stimulus; and whether they watched the demonstrator, defined as looking in the direction of the demonstrator while stationary, or not, which was defined as not looking in the direction of the demonstrator, or glancing at the demonstrator but not stopping to watch, while at this proximity. The difference in dominance rank between each demonstrator and observer was also recorded.

We encountered two unanticipated difficulties with this series of experiments. First, subordinates occasionally moved away carrying the stimulus when they were approached by dominants. In these cases, the experimenter followed the baboon with the stimulus but did not record the ‘watching’ durations of the observers, as these were unlikely to be accurate. Second, because individuals moved with the task through the troop as it was foraging, it was impossible to present the task to particular individuals in a particular order. Individuals were thus sometimes able to experience a ‘learn’ trial before a ‘test’ trial, and we were unable to systematically achieve the more controlled test—learn—re-test experiments of the novel food task. Further, we were unable to control the number of times an individual was tested with or without a demonstrator. However, this situation is more likely to reflect the natural opportunities for social learning than our more controlled test—learn—re-test paradigm, and we have therefore included this series of experiments in our analyses. In this case, for each individual, we recorded three possible trial presentation numbers that reflect the different types of information that they could acquire: the cumulative presentation number (cumulative total social and personal information), the number of presentations with a demonstrator (cumulative social information) and the number of presentations without a demonstrator (cumulative personal information).

Because we retrieved a task after it had been solved, no baboons could have had a ‘learn’ presentation (with a demonstrator) and a ‘re-test’ presentation (without a demonstrator) in the same trial. That is, no individuals could watch a demonstrator and then subsequently interact with an unsolved task, though some individuals did interact with parts of the task that were ripped off and dropped by the demonstrator before the experimenter could retrieve them. Individual baboons experienced the task with a demonstrator a median of 3 times (inter-quartile range (IQR) = 2–5, range = 1–11 times) and without a demonstrator a median of 2 times (IQR = 1–3, range = 1–9 times), and in total (either with a demonstrator or without) a median of 5 times (IQR = 1–3, range = 1–16 times) with a mean interval of 3.30 ± 0.47 days (range = 0–52 days) between trials. We performed a total of 243 baboon-trials to 45 baboons with 18 demonstrators who were demonstrators for a median of 5.5 individuals (IQR = 4–15.5, range = 1–25 individuals). The demonstrators for this experiment comprised 11 male and 6 females (and one infant) who were on average 6 years old, had a relative rank of 0.64, with an average boldness of 0.33 and an average anxiety of 0.06. The focal subjects for this experiment comprised 19 male and 22 female baboons who were on average 7.5 years old, had a relative rank of 0.54, with an average boldness of 0.44 and an average anxiety of −0.03.

We confirm that we have adhered to the Guidelines for the Use of Animal Behaviour for Research and Teaching (2003). Our experimental protocols were assessed and approved by the Ethics Committee of the Zoological Society of London (BPE/052). Our study was approved by the Ministry of Environment and Tourism in Namibia (Research/Collecting Permits 1379/2009, 1486/2010, and 1486/2011).

Statistical analyses

General modelling approach

All data were analysed using (generalised) linear mixed effects models (LMEs or GLMMs), fitted in the R environment (R Development Core Team, 2011). Unless otherwise specified below, all models started with all fixed and random effects, and fixed effects were sequentially dropped (from least significant terms first) until a minimal model was obtained that included only those effects which explained a significant amount of the variation in the model. Each dropped term was added to the minimal (final) model to check that it remained non-significant. Significance of terms was determined by p-values (all effects with a p-value ≤ 0.05 were retained in the models). Full models included the following fixed effects: boldness, anxiety, sex (factor: male or female), age class (factor: juvenile or adult), and rank (continuous variable: standardised between 0 and 1 within each troop). We also included the rank difference between the test subject and the demonstrator in those models investigating whether or not the subject watched the demonstrator (continuous variable; see models 2 described below). Finally, we included presentation number as a fixed effect in all hidden food task models as individuals were exposed to the stimuli on multiple occasions in this experiment. In all models, we included test subject identity and troop identity (to control for repeated measures) as random effects. The latter was further crossed with demonstrator identity in those models investigating whether or not the test subjects watched a demonstrator (models 2). All Gaussian models’ residuals were normally distributed, and there were no covariances ≥0.40 between fixed effects.

Novel food experiments

We did not analyse which individuals initially solved the novel food task (Q1), since the processing and consumption of the novel food was the basis of our boldness assay. We thus first investigated whether individuals of different personality types spent longer watching a demonstrator (Q2, predictions 2a and b, i.e., P2NFa and b). Using data obtained from the novel food experiments when a demonstrator was interacting with the food (n = 54 baboon-trials with 36 baboons observing 14 demonstrators), we analysed the time the baboons spent watching the demonstrator (s) with a Poisson error structure (model 2, novel food: m2NF).

Next (Q3), we investigated whether individuals of different personality types were more or less likely to change their behaviour, i.e., learn, after being exposed to social information via a demonstrator (predictions 3a and b, i.e., P3NFa and b). We investigated whether the baboons were more likely to increase their handling time of a novel food after watching a demonstrator, and whether this depended on boldness or anxiety, by investigating whether there was a three-way interaction between the personality trait, the presentation (‘test’ or ‘re-test’), and the treatment (control or treatment) (models 3NF; control: n = 58 paired presentations to 23 baboons over 2009–2011, treatment: n = 32 paired presentations to 16 baboons in 2010). However, as we could not concurrently investigate a three-way interaction between both boldness and anxiety due to overparameterization, we performed two models, one for each of the personality traits (m3NFa and b).

Hidden food experiments

First (Q1), we investigated whether personality predicted whether individuals found food in the hidden food experiment (i.e., solved the task; see predictions P1HFa and b). We analysed whether an individual managed to solve the hidden-food task if it had an opportunity to do so (i.e., obtain food from it, binary 1/0; n = 80 occasions when a baboon interacted with the tetrapack, involving 33 different baboons) using a GLMM with a binomial error structure (model 1, hidden food experiment: m1HF). As previously mentioned, few individuals had the opportunity to interact with the task without previously having seen a demonstrator interacting with the task; consequently, this analysis cannot distinguish between task-solving due to ‘innovation’ or social learning (but see Results).

Next, we investigated whether individuals of different personality types were more likely to watch a demonstrator (Q2, predictions P2HFa and b) using data obtained from the hidden-food experiments with a demonstrator. We analysed whether individuals watched the demonstrator or not (n = 163 baboon-trials with 41 baboons observing 14 demonstrators), using a binomial error structure (model 2HF).

Finally, we investigated whether the baboons were more likely to increase their response to the hidden-food stimulus after watching a demonstrator, and whether this depended on boldness or anxiety by investigating whether there was an interaction between either of these traits and the presentation number (Q3, predictions P3HFa and b). In this case, we categorised the subjects’ responses on a five-point scale, coded 0–4 in order of increasing response to the stimulus, as: ignore, the individual walked within 5 m of the stimulus without showing a response; look, the individual walked or stopped within 5 m of the stimulus and looked towards it; smell, the individual approached the stimulus and bent to smell it; handle, the individual approached the stimulus and picked it up; or eat, the individual obtained food from the stimulus. As the models’ residuals were normally distributed when analysed with a Gaussian error, we used linear mixed effects models to model the data (models 3HF) rather than generalised linear mixed models for ordinal data. Due to the design of the hidden food task, subjects varied in the number of presentations they had experienced. To allow for this variation, we ran three models with different predictors: the cumulative number of all presentations (total exposure), the cumulative number of ‘test’ presentations without a demonstrator, and the cumulative number of ‘learn’ presentations with a demonstrator. We used backward model selection on all three models, each including one of the different cumulative presentation numbers (total, personal and social information presentations) and then compared with a log likelihood ratio test which of the three minimal models best explained the data, which included 80 baboon-trials to 33 individuals.

Results

The minimal models for all analyses described below are provided in Table 1, and a summary of our findings with regard to our predictions is provided in Table 2. In addition to the results we present below regarding personality, it is notable that age class had an effect on many of the variables that we investigated (Table 1). In all cases, there was a positive effect of being juvenile on our measured variables: juveniles solved the hidden task more often (m1HF), spent longer watching a demonstrator (m2NF), and showed evidence for greater social learning than adults: juveniles handled the novel food for longer after watching a demonstrator (m3NFa), and showed greater responses to (and improvement in, see File S1) the hidden food task (m3HF). Few other variables had such a consistent effect on our measurements (Table 1).

Table 2 The questions tested, experiments conducted, data and models used to test the predictions, and whether we found support for our predictions or not (outlined briefly here).

Predictions highlighted in bold show that we found support for these in our models. Note that this may indicate that there was not a significant relationship. Predictions for which we did not find support but found the opposite trend are in bold and italicized. ‘N/A’ is listed for both boldness and anxiety effects in the novel food experiment under Question 1. This is because, in the case of boldness, this experiment was also our assay for boldness, and in the case of anxiety, previous work has demonstrated that anxiety is unrelated to boldness (and hence performance in the novel food experiment) in this species (Carter et al., 2012c).

					Prediction and whether it was supported	
Question	Test	Experiment	Data analysed	Model	Anxiety	Boldness	
1	Propensity to solve the task	Novel food	N/A		N/A	N/A	
		Hidden food	Whether the baboon ate food from the task or not (0/1)	m1	P1a: calm-anxious animals unlikely to differ
YES	P1b: bold animals are likely to solve
YES	
2	Watching a demonstrator	Novel food	Time (s) spent watching a demonstrator	m2nf	P2a: anxious animals will be more attentive
No, calmer individuals
were more attentive	P2b: shy animals will be more attentive
NO (neither)	
		Hidden food	Whether the baboon watched a demonstrator or not (0/1)	m2hf	P2a: anxious animals will be more attentive
No (neither)	P2b: shy animals will be more attentive
NO (neither)	
3	Change in task-solving behaviour after watching a demonstrator	Novel food	Time spent handling the novel food (s) after treatment/control	m3nf	P3a: anxious animals will show greater improvement
YES	P3b: shy animals will show greater improvement
NO, bold
individuals improved.	
		Hidden food	Level of interaction with the hidden food container
(levels 0–4)	Model set 3hf	P3a: anxious animals will show greater improvement
NO (neither)	P3b: shy animals will show greater improvement
NO (neither)	

Novel food experiments

We assessed individual variation in the time spent observing demonstrators manipulate the novel food (Q2). We found no evidence that shyer baboons were more attentive observers, contrary to P2NFb. Further, in contrast to P2NFa, calmer baboons spent more time watching demonstrators interacting with a novel food, after accounting for the subject’s age and rank (m2NF: β ± s.e. = −2.79 ± 0.51, t = −2.01, Fig. 1).

Figure 1 The times that individuals spent watching a demonstrator manipulate a novel food.

The average time anxious and calm juvenile and adult baboons spent watching a demonstrator (s) interact with a novel food (n = 54 trials). Plotted are the means and standard errors for the raw data; note that though the data are presented as categories (split at the mean of the trait for the population) they were analysed as continuous variables.

Next, we assessed the evidence for social learning, and individual variation in social learning arising from personality differences. In the case of novel foods, we achieved this by analysing the food handling time in the ‘re-test’ conditions both for those baboons that either had no exposure to a demonstrator after the initial test (the control group) and for those that did (the experimental group). In both the models for anxiety (a) and boldness (b) (models 3NFa and b), the three-way interaction was significant (3NFa: β ± s.e. = 7.66 ± 3.04, z = 2.52, p = 0.01; 3NFb: β ± s.e. = −0.68 ± 0.08, z = −8.67, p < 0.001). Overall, in the control groups, individuals spent less time handling the novel food when re-tested, and both bolder and more anxious individuals showed a greater decline in handling time when re-tested (Table 1; Figs. 2A and 2C). In the treatment groups, individuals showed the opposite response—handling times tended to increase when re-tested—and bolder and more anxious individuals showed higher handling times than shyer and calmer individuals, respectively (models 3NFa and b, Table 1; Figs. 2B and 2D). Although calmer individuals also showed higher handling times in the second presentation after watching a demonstrator, this reflected a non-significant 1.4-fold increase in handling time in comparison to the 2.7-fold increase in more anxious individuals. Thus, bolder and more anxious animals showed social learning after exposure to a demonstrator (contrary to P3NFb, but in support of P3NFa, respectively).

Figure 2 Individual responses to a novel food after an opportunity for social learning.

The average time that bold and shy (A, B) and anxious and calm (C, D) baboons handled the novel food on their first interaction with it (pres. 1) and on their second interaction with it (pres. 2) either without being exposed to a demonstrator (A, C) or after being exposed to a demonstrator (B, D) (n = 45 trials). Plotted are the means and standard errors for the raw data; note that though the data are presented as categories (split at the mean of the trait for the population) they were analysed as continuous variables.

Hidden food experiments

In our assessment of individual propensity for task-solving (Q1), we found that bolder baboons were more likely to obtain food from the hidden food experiment (β ± s.e. = −0.85 ± 0.32, z = −2.63, p = 0.008), after accounting for differences in age (model 1, Table 1; Fig. 3), in support of P1HFb. Anxiety, however, had no effect on task-solving, in support of P1HFa (model 1: β ± s.e. = −0.28 ± 0.72, z = −0.40, p = 0.69).

Figure 3 The proportions of individuals that solved the task.

Differences in proportions of adult and juvenile bold and shy baboons that ‘solved’ the hidden food task by obtaining and eating food from the box (n = 80). Plotted are the means and standard errors for the raw data; note that though the data are presented as categories (split at the mean of the trait for the population) they were analysed as continuous variables.

We then assessed individual variation in the probability of observing the demonstrators (Q2). We found no evidence that shyer or more anxious baboons were more likely to watch an individual manipulate the hidden food task, contrary to P2HFb and a respectively (Table 1, m2HF).

Finally, we assessed the evidence for social learning by analysing the intensity of the response to the stimulus in relation to the number of prior total, social or personal presentations. We found no evidence for an interaction between any of the presentation numbers and either personality trait; in all cases the minimal model was the same (model 3HF). Thus, while bolder animals continued to be better at solving the hidden food task (as observed in m1), we found no evidence that either shyer or more anxious individuals showed improved responses with more prior experience, contrary to predictions 3HFb and 3HFa respectively.

Discussion

Our results strongly suggest that personality predicts social learning propensity in wild baboons. In a field experiment in which individuals had the opportunity to learn about a novel food from an experienced demonstrator, both bolder and more anxious individuals increased their handling time of the novel food if they had had an opportunity for social learning. Below we address those predictions that were not supported before focussing on three important issues emerging from our study that have implications not only for understanding social learning in the wild (van de Waal & Bshary, 2011), but also for individual differences in cognitive abilities (Thornton & Lukas, 2012) and the formation of traditions and culture in animal groups (Whiten, 2000).

Contrary to expectation, we found no effect of boldness on time spent watching a demonstrator. These results differ from those of Kurvers et al. (2010) which suggested that shy individuals used social information more than bold individuals, but this discrepancy may be explained by differences between studies in the familiarity of the study subjects with the experimental challenge (see below). However, our finding that bold individuals were more likely to solve the hidden food task confirms previous research showing an increase in task-solving success when individuals are less neophobic (Seferta et al., 2001; Webster & Lefebvre, 2001). Our failure to find a comparable improvement in task solving after watching a demonstrator for the hidden food experiment is unexpected, but may reflect the fact that the task contained a highly preferred food, and individuals may be limited by their rank in their access to the task. That is, individuals may have acquired the social information necessary to solve the task, but were unable or unwilling to access the task due to their lower rank. Though we do not have the resolution of data to test this explicitly, we discuss this possibility further below.

Three important issues emerged from our study. First, an individual’s ability or interest in collecting social information does not necessarily correlate with its ability or interest in using social information. This observation arises from our findings that bolder and more anxious individuals showed a greater improvement than shyer and calmer individuals in their response to a novel food after watching a demonstrator, despite the fact that there was no effect of boldness on the time spent watching a demonstrator, and calmer individuals were more attentive than more anxious individuals. Three possible explanations could be considered for these results: that (i) a mechanism other than watching a demonstrator is responsible for social learning in baboons, (ii) baboons need little time to acquire sufficient social information to subsequently change their behaviour, or (iii) as mentioned above, the acquisition of social information is not always correlated with its use. The first explanation is unlikely, since the task is a visual one and baboons are strongly visual animals. It therefore seems probable that the second or third explanations are most likely, and these need not be mutually exclusive.

Social learning mechanisms such as stimulus enhancement, where the behaviour or presence of a demonstrator attracts the attention of a naïve individual (Brown & Laland, 2003), may quickly result in a behavioural change in naïve individuals without the detailed and lengthy observation assumed necessary in processes such as imitation. As such, the baboons could learn that an object is worth investigating in more detail in the future in the time it takes to identify the object that a demonstrator is manipulating, as may be the case in our study. Further, it may be that bold individuals show greater social learning not necessarily because they are ‘better’ at collecting or processing social information, but because they are generally more willing to interact with a novel task. This, coupled with the rapid social learning through stimulus enhancement we posit to be possible in this study, may result in bolder individuals showing greater social learning. In support of this interpretation, in the novel food control trials, bold baboons decreased their handling time of the novel food but this handling time still exceeded that of shy individuals on their first presentations. This may also explain why our findings differ to those of Kurvers et al. (2010), as the authors of that study habituated the study subjects to the experimental arena prior to testing. In both cases (our study, and Kurvers et al.’s study), individual variation in response to novel tasks is related to personality, but not necessarily an individual’s ability to collect and process social information. The fact that some individuals may be too shy or anxious to interact with a novel task, despite having the information or ability to do so, may have wide implications. For example, an individual’s cognitive performance may be systematically underestimated if it is unwilling, but still able, to solve a task, which may hinder studies interested in the fitness advantages and evolution of cognitive performance. Future studies could assess this effect experimentally by testing study animals of different boldness with varying degrees of habituation to a test apparatus. Our interpretation would lead to the prediction that bolder individuals should outperform shyer individuals under low habituation regimes, but that performance may be more equal when the task is no longer threatening.

Second, the identity of demonstrators may be important. In both our experiments, many of our initial demonstrators were bold. In the novel food experiment, this was inevitable, as our definition of boldness precluded individuals that did not eat the novel food. However, it also transpired that those individuals that solved the hidden food task were more likely to be bold, though we note that the boldness and anxiety of the demonstrators was, overall, not particularly different to the boldness and anxiety of the population (see summary statistics reported in the methods for these groups). As these experiments were not designed to address this particular question and thus the sample of demonstrators is limited, we have not analysed these data here. However, previous research has suggested that the characteristics of a demonstrator may affect whether an observer gathers or uses social information from them or not (termed ‘directed social learning’: (Coussi-Korbel & Fragaszy, 1995; Laland, 2004)). For instance, wild vervet monkeys (Chlorocebus aethiops) are more likely to attend to the social information provided by the philopatric sex (females) (van de Waal et al., 2010). It remains to be established whether the personality of the demonstrator might be important, but it seems possible that it may play a role. For example, slow explorers are more thorough at exploring novel environments (Careau et al., 2009) and may therefore be better informed about the location of resources. Although it is difficult to anticipate how demonstrator boldness might affect audience attentiveness and social learning in baboons, we can consider its correlates. Specifically, while sex is unrelated to boldness in our population, there is an effect of age: juvenile baboons are bolder than adults (A Carter, H Marshall, G Cowlishaw, unpublished data, 2009–2011). Since this led to a strong representation of juvenile animals among the demonstrators (10 juvenile, 4 adult demonstrators), and adults may attend less closely to juveniles (only vertical transmission of social information has been documented in this species: Camberfort, 1981), it is possible that the adults in our sample might have shown lower levels of attentiveness and social learning than would have been observed with adult-only demonstrators.

Finally, an important aspect of our study that we did not anticipate was the difficulty of testing certain individuals in the presence of a demonstrator. While we endeavoured to provide all individuals with the opportunity to watch a demonstrator for both tasks, it was not possible to find a situation in which certain individuals were foraging in sufficiently close proximity to potential demonstrators for this to occur (Sih & Del Giudice, 2012). In great tits (Parus major), slow exploring birds showed higher attraction to unknown conspecifics than fast explorers (Carere et al., 2001) suggesting that personality may influence sociability and thus opportunities to learn from others in some species. Further, in guppies (Poecilia reticulata), an individual’s social network is related to its personality: individuals associated assortatively according to behavioural type, and shy fish had more and stronger network connections than bold fish (Croft et al., 2009). An individual’s social network is thus likely to affect which individuals can supply social information, and thus the flow of information through a group (Voelkl & Noë, 2008; Sih, Hanser & McHugh, 2009; Claidiére et al., 2013). If similar personality-related patterning among the social networks of baboons occurs, it may be impossible for some individuals to obtain social information about a novel food or task as they may have only weak associations with demonstrators. Together with the preceding points, it thus appears possible that personality could affect social learning, and hence the formation of culture in animal groups, through a combination of mechanisms. These include not only the differential collection of social information, as we have shown in this study, but also the differential application of social information, directed social learning, and assortative social bonds.

Supplemental information

Video S1 Social learning experiments

The video shows (1) an individual extracting food from the hidden food task, (2) individuals observing the demonstrator (to the right of the screen), (3) an infant (data for which were not recorded) showing anxiety towards a hidden food task, (4) a juvenile individual (demonstrator) interacting with the novel food task while an adult does not watch (look in the direction of the demonstrator) and (5) a juvenile observer watching the demonstrator.

Click here for additional data file.

File S1 Individual responses to hidden food task

Factors affecting social learning of the hidden food task when an interaction between the presentation number and age class is considered.

Click here for additional data file.

Supplemental Information 3 Data for analysis m1HF

“Baboon”, integer; identity code for the focal individual “Solve”, binary; whether the baboon solved the task (1) or not (2) “Presentation”, integer; the presentation order “Sex”, M = male, F = female “RelRank”, the relative rank of the individual between 0 (lowest rank) and 1 (highest rank) “AgeClass”, the age class of the individual: juvenile or adult “Boldness”, the boldness score of the individual “Anxiety”, the anxiety score of the individual “Troop”, the troop to which the baboon belongs: J or L.

Click here for additional data file.

Supplemental Information 4 Data for analysis m2NF

“Observer”, integer; identity code for the focal individual “Demonstrator”, integer; identity code for the demonstrator “Presentation”, integer; the presentation order “TimeWatching”, integer; the time (s) the focal individual spent watching the demonstrator manipulate the novel food “Sex”, M = male, F = female “RelRank”, the relative rank of the individual between 0 (lowest rank) and 1 (highest rank) “AgeClass”, the age class of the individual: juvenile or adult “RankDiffDemObs”, the difference between the relative ranks of the demonstrator and the observer “Boldness”, the boldness score of the individual “Anxiety”, the anxiety score of the individual “Troop”, the troop to which the baboon belongs: J or L.

Click here for additional data file.

Supplemental Information 5 Data for analysis m2HF

“Baboon”, integer; identity code for the focal individual “Demonstrator”, integer; identity code for the demonstrator “WatchOrNot”, binary; whether the focal individual watched the demonstrator (1) or not (0) “Presentation”, integer; the presentation order “Sex”, M = male, F = female “RelRank”, the relative rank of the individual between 0 (lowest rank) and 1 (highest rank) “AgeClass”, the age class of the individual: juvenile or adult “RankDiffDemObs”, the difference between the relative ranks of the demonstrator and the observer “Boldness”, the boldness score of the individual “Anxiety”, the anxiety score of the individual “Troop”, the troop to which the baboon belongs: J or L.

Click here for additional data file.

Supplemental Information 6 Data for analysis m3NF

“Baboon”, integer; identity code for the focal individual “Troop”, the troop to which the baboon belongs: J or L “Sex”, M = male, F = female “AgeClass”, the age class of the individual: juvenile or adult “RelRank”, the relative rank of the individual between 0 (lowest rank) and 1 (highest rank) “HandleTime”, the time spent handling the novel food (s) “pres”, integer; the presentation order “Boldness”, the boldness score of the individual “Anxiety”, the anxiety score of the individual “Exp”, the experiment the baboon was involved in, either the treatment (Treat) or control (Control).

Click here for additional data file.

Supplemental Information 7 Data for analysis m3HF

“Baboon”, integer; identity code for the focal individual “Presentation”, integer; the cumulative total information presentation order “Troop”, the troop to which the baboon belongs: J or L “Sex”, M = male, F = female “RelRank”, the relative rank of the individual between 0 (lowest rank) and 1 (highest rank) “AgeClass”, the age class of the individual: juvenile or adult “Boldness”, the boldness score of the individual “Anxiety”, the anxiety score of the individual “Troop”, the troop to which the baboon belongs: J or L “InteractionLevels”, integer; the level of interaction the baboon had with the hidden food task (0–4, see main text for details). “PersonalInfoAcquired”, integer; the cumulative personal information presentation order “SocialInfoAcquiredToPres” integer; the cumulative social information presentation order.

Click here for additional data file.

We thank Baboon Teams 2009, 2010 and 2011 for putting up with AJC standing around and swearing at baboons a lot, especially Claudia Sick who performed some of the novel food control presentations in 2009; Drandrew Bateman, Corina Logan and Dieter Lukas for commenting on an earlier version of the manuscript; and Pierre-Olivier Montiglio, Alex Weiss and an anonymous reviewer for helpful reviews. We are grateful to the Ministry of Lands and Resettlement for permission to work at Tsaobis Leopard Park, the Gobabeb Training and Research Centre for affiliation, and the Ministry of Environment and Tourism for research permission in Namibia. We are also grateful to the Snyman and Wittreich families for permission to work on neighbouring farms. This paper is a publication of the ZSL Institute of Zoology’s Tsaobis Baboon Project.

Additional Information and Declarations

Competing Interests

Author Contributions

Animal Ethics

Field Study Permissions

The authors declare they have no competing interests.

Alecia J. Carter conceived and designed the experiments, performed the experiments, analyzed the data, contributed reagents/materials/analysis tools, wrote the paper, prepared figures and/or tables, reviewed drafts of the paper.

Harry H. Marshall performed the experiments, wrote the paper, reviewed drafts of the paper.

Robert Heinsohn and Guy Cowlishaw wrote the paper, reviewed drafts of the paper.

The following information was supplied relating to ethical approvals (i.e., approving body and any reference numbers):

We adhered to the Guidelines for the Use of Animal Behaviour for Research and Teaching (2003). Our experimental protocols were assessed and approved by the Ethics Committee of the Zoological Society of London (BPE/052).

The following information was supplied relating to ethical approvals (i.e., approving body and any reference numbers):

Our study was approved by the Ministry of Environment and Tourism in Namibia (Research/Collecting Permits 1379/2009, 1486/2010, and 1486/2011).

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
