# Peer review of "Personality predicts the propensity for social learning in a wild primate"

_PeerJ, doi:10.7717/peerj.283_

## Round 0.1 · original submission · Major Revisions

We have now received three reviews on your manuscripts. All reviewers found merits in your study, but also raised important points that deserve further attention. First, relevant literature needs to be integrated to the manuscript (the reviewers made a few suggestions in that respect but a thorough review of the literature should also be performed). Second, all reviewers emphasized that the categorization of the individuals should be better explained, as it is confusing as it stands. Also, the predictions and hypotheses should be clarified in a single section at the end of the introduction. I also agree with reviewer 2 that a clearer presentation of the results could be achieved by using a more 'biological' and less 'statistical' description. Finally, the reviewers provided several additional points that need to be assessed in your revision to further improve your manuscript. As such, you should present a detailed cover letter that will describe your response to every comment.

Reviewer 1 ·

Basic reporting

This is a fairly exhaustive study on an extremely interesting and timely topic. To my knowledge the link between personality and social cognition has never been so exhaustively investigated as in this study.

Experimental design

No comments

Validity of the findings

The findings seem very important and in partial agreement with one similar study. They may have important implications as they posit a role of personality not only in ability in acquisition of information, but also in its use. In this respect, when (temporally) and under which circumstances and contexts is then used is also a crucial variable, not assessed here. However, the categorization of the individuals should be better explained to judge the validity of the findings

Additional comments

Review of Personality predicts the propensity for social learning in a wild primate” by Carter et al..

This paper describes the association between anxiety, boldness and social learning performance in two groups of baboons in the wild. Such groups have been already characterized for personality and two papers just published. This is a fairly exhaustive study on an extremely interesting and timely topic. To my knowledge the link between personality and social cognition has been never so exhaustively investigated as in this study. In a recent compilation on Animal Personality by Carere and Maestripieri, no data on this potential implication of personality is reported apart from some possible predictions as avenue for future research.

The paper is very well written and presented, with well delineated specific questions, but seems to lack in clarity for some of the measures and analyses. Apart for some important recent literature that should be screened and where appropriate incorporated, a major unclear point that leaves the reader unsatisfied is the relationship between bold, shy, anxious and calm in the studied individuals. I guess that things may become more clear if one would read the two papers of the same author on the same animals, but still something should be explained here: are the categories anxious and calm nested into bold and shy? A bold individual can be either anxious or calm? Or is it the other way around? Or are they all mutually exclusive? Unravelling these relationships is not trivial, also because one major issue in the recent surge of studies is whether personalities cope differently with stress and anxiety and this could in turn affect cognitive performance. As it is presented, the categorization may seem an oversimplification.

Minor points

Abstract
I suggest mentioning the name of the studied species in the text

Introduction
Lines 20-23 I suppose there are studies highlighting potentil or actul costs of social learning. One obvious is that information may be not based on own cues/control directly and may lead to errors. It would be good if also potential disadvantages are outlined, which may explain also the evolution/adoption of lower propensity.
Line 30 You may also cite the recent volume on Animal Personality to be more updated and inform the general reader of the relevance of the topic. Also in the following part you may screen and cite the conceptual reviews by Sih and Del Giudice and by Carere & Locurto.
Lines 40-44 I think propensity to social exploration might play a role. As far as I know, there is some empirical evidence that personality differ in social exploration in great tit (Carere et al. 2001), but it is worthy to take the occasion to better explore the literature.

Methods
Line 89 Why do you call anxiety the response to a predator? It is fine to me, maybe this normal terminology in primates.
The personality assessments section needs more explanation (see general comment)

Discussion
A good discussion overall
I would suggest starting more modestly with something like “Our results strongly suggest that…”
Lines 365-367. Indeed this seems to have important implications, but could tell a bit more about what sorts of implications?


Suggested references
Carere C, Welink D, Drent PJ, Koolhaas JM Groothuis TGG 2001 Effect of social defeat in a territorial bird (Parus major) selected for different coping styles. Physiol Behav 73, 427-433
Carere C, Locurto J 2011 Interaction between animal personalities and animal cognition. Curr Zool 57, 491-498
Carere C, Maestripieri D (Eds.). 2013 Animal Personalities: Behavior, Physiology, and Evolution. The University of Chicago Press, Chicago, London. (see especially chapter by Weiss & Adams Differential Behavioral Ecology: The Structure, Life History, and Evolution of Primate Personality)
Sih A, Del Giudice M 2012 Linking behavioural syndromes and cognition: a behavioural ecology perspective Philos Trans R Soc Lond B Biol Sci 367, 2762-2772

·

Basic reporting

The submission must adhere to all PeerJ policies (see: 'Journal Policies').
R : To the extent of my knowledge, it does.

The article must be written in English using clear and unambiguous text and must conform to professional standards of courtesy and expression.
R : Overall, the manuscript is very well written and clear (especially the introduction and discussion). The rational presented in the introduction follows a very clear line of though. The structure of the discussion is also logical and easy to follow. A few points in the presentation could be improved. First the paper uses a fair amount of technical terminology (boldness, anxiety, personality, social learning). It is not always obvious why this terminology is necessary. I outline where this is the case in the main and line by line comments. I also suggested some ways to improve this issue. Second, the predictions and hypotheses are scattered in more than one place in the manuscript. These are briefly (too briefly) in the introduction, detailed a little in the methods and summarized in a table. I suggest developing the introduction to include a clear presentation of all the hypotheses and predictions there. Third, the presentation of the experimental design and of the results are lacking some background information. Fourth, the statistical analyses in the methods section and the results section could follow a more linear structure, presenting each of the three phases of the experimental design one after the other (test, learn, re-test).

The article should include sufficient introduction and background to demonstrate how the work fits into the broader field of knowledge. Relevant prior literature should be appropriately referenced.
R : As I outlined in my comments the the authors, I think the rationale is sound and clear. The background seems sufficient, but could eventually include the recent theoretical work by M. Wolf and others, on how personality and social learning should interact. Presenting this literature will help the authors make the point that their study is important because we are lacking empirical investigations on the topic. More background is needed on the previous studies on the study system. This is especially necessary in the methods.

The structure of the submitted article should conform to an acceptable format of ‘standard sections’ (see our Instructions for Authors for our suggested format). Significant departures in structure should be made only if they significantly improve clarity or conform to a discipline-specific custom.
R : The general structure is adequate.

Figures should be relevant to the content of the article, of sufficient resolution, and appropriately described and labeled.
R : The authors need to add sample sizes and clearly present what the error bars represent on all figures. Figures are clear. Figures currently appear pixelized but this may be due to their large size. Figure 3 should clearly present which panels are treatment and which are controls.

The submission should be ‘self-contained,’ should represent an appropriate ‘unit of publication’, and should include all results relevant to the hypothesis. Coherent bodies of work should not be inappropriately subdivided merely to increase publication count.
R : This publication is a solid and coherent unit, although I suggest the put more emphasis on some aspects of the work.

Experimental design

The submission must describe original primary research within the Aims & Scope of the Journal. This study presents an original, empirical investigation in the biological sciences.
R : It fits within the Aims and scope of PeerJ.

The submission should clearly define the research question, which must be relevant and meaningful. The research question is clearly defined, highly relevant.
R : The authors make testable predictions.

The investigation must have been conducted rigorously and to a high technical standard.
R : The authors present a rigorous test of their hypotheses. At the same time, they also acknowledge and discuss the shortcomings associated with the nature of the study system.

Methods should be described with sufficient information to be reproducible by another investigator.
R : More background information should be given on the behavioral tests used to quantify boldness and anxiety. I outlined the missing information below in the main comments.

The research must have been conducted in conformity with the prevailing ethical standards in the field.
R : The authors explicitly present their guidelines at lines 265 to 270. The experimental design is unlikely to pose any ethical concerns.

Validity of the findings

The data should be robust, statistically sound, and controlled.
R : They are.

The data on which the conclusions are based must be provided or made available in an acceptable discipline-specific repository.
R : The authors do not mention if their data will be made available publicly.

The conclusions should be appropriately stated, should be connected to the original question investigated, and should be limited to those supported by the results.
R : The conclusions are logical interpretations of the results.

Speculation is welcomed, but should be identified as such.
R : The authors make a clear distinction between their results and their speculations. Overall, speculations could be a little bit more specific. For example, what would be an alternative mechanism for the acquisition of social learning (other than watching co-specifics)? See also comments to the authors.

Decisions are not made based on any subjective determination of impact, degree of advance, novelty, being of interest to only a niche audience, etc.
R : The experiments and analyses presented in this study deserve publication.

Replication experiments are encouraged (provided the rationale for the replication is clearly described); however, we do not allow the ‘pointless’ repetition of well known, widely accepted results.
R : This manuscript presents an original and novel study.

Additional comments

Overall the study presents an interesting, and robust empirical investigation of the links between individual behavioral variation (termed personality) and the propensity to use social learning to sample the environment. This manuscript presents result on two experiments carried on two groups of baboons to investigate how an individual's behavioral response toward novelty affects its ability to acquire and use social information. The individuals were presented with a novel food item individually, allowed to acquire social information by watching a co specific interact with the food (treatment) or simply watch the food item (control) and were retested individually to measure their response to the food item afterwards. An additional experiment used a similar approach, but used a food item hidden in a container instead. The authors found that the initial reaction of individuals towards the novel food item (when tested individually) predicted their tendency to acquire social information, and their change in response between the test and re-test phases of the experiments. Their main conclusion is that an individual's behavioral response toward novelty affects their ability to acquire social information. Another important point is that social information acquisition is not always associated with social information use.

The rationale is clear, the experimental design and statistical analyses are robust. The conclusions appear as a logical interpretation of the results. I congratulate the authors on their experimental design, which elegantly deals with the (many) limitations imposed by the study system. I make a series of comments below (see also the three previous sections of the review). I hope these will help the authors improve their interesting manuscript. Most of my comments either ask for more information on the experimental design and study system, or point to aspects of the work that potentially deserve more emphasis. The manuscript is suited for publication if the authors provide the additional information I outline.


Pierre-Olivier Montiglio


Comments:
The labels used for both personality traits are somewhat counter intuitive, For example, boldness, usually described as the willingness to take risk, could apply to both assays. However, most studies have investigated the behavioral response to an increase predation risk. Similarly, tendency to approach food, has been traditionally described as boldness also. Anxiety is a less well defined term. For example is it usually associated with increased physiological stress reactivity. Is it the case in this system? The authors should provide more background on the assays and personality traits : what is their repeatability, how many individuals and how many times were individuals assayed, are the two traits associated among individuals? Is there any knowledge on whether shy/anxious individuals differ in dominance from bold/calm ones? How are these traits changing with age? Clarifications on the labels used to describe personality could be provided briefly at lines 57-58, and the more technical details at lines 85 and following. Parts of the 'social learning' section could also be moved up to the 'personality assessment' section to help clarify.

Also, the authors classify the individuals in dichotomous shy-bold and anxious-clam classes. While I understand the authors justification, I think that this classification may mask some inconsistencies. At this point in the manuscript, the reader is lead to think the assays provide a continuous assessment of each personality trait. If this is the case, this is a pity to 'throw' away a portion of the information available to enable comparisons. I suggest that the authors present analyses using the continuous data (if such data is available). The results using dichotomous classes could always be provided as supplementary material to facilitate discussion of the results. This should also enable the reader to determine for himself if the dichotomous classification is adequate. Part of this issue may be coming from the scarce background provided for the tests (see previous paragraph).

Boldness is investigated using the same assay as social learning (in fact both are measure within the same experiment). This may seem a bit circular at first, and overall leads the reader to think that the terminology (boldness, anxiety, social learning) is not describing qualitatively different phenomena. Additional justifications/explanations could also be provided to show that there is no real flaw in the novel food experiment. A crucial point is whether bold and calm individuals are constrained in their ability to use social information, because they already expressed the maximum value during the 'test' phase. Could the bold individuals increase their tendency to approach food from the test to the retest? Also, since the authors did not control which individual received a control or a demonstration, is there a potential bias in personality between control and treatment groups. Please provide details to justify why this is not likely to affect the validity of the results. I do not think this a problem here, and this shortcoming does not invalidate the study. However, I suggest simplifying the terminology to make the story clearer and more direct. Rather than presenting the study as a test of the relationship between boldness and social learning, it would make more sense to present it as one experiment assessing tendency to approach food and how it constraints/predicts the use of social information in subsequent encounters with that food.

Presentation of the hypotheses and prediction could be made clearer by presenting them in the introduction. This will encourage the introduction to be more specific and precise as to what the authors are testing. The introduction is already clear and follows a logical path of reasoning. Making it more precise and linking it tightly to the hypotheses and prediction will make it even more effective. In addition, having the precise hypotheses and predictions in mind will simplify the reader's work when reading the methods. Similarly, presentation of the statistical analyses are somewhat confusing because they require the reader to 'jump' between the first paragraph of the section (describing the general approach and the independent variables used), each question (including the dependent variables) and a table. The authors also refers to multiple labels for each models (i.e. m1, m2HF, m3HFa vs b, etc). I suggest restructuring this information along the three questions (which is also the structure of the result section), presenting the models for each question. This may lead to repeat some of the information, but will make the analyses much clearer.

Clearer presentation of the results could be achieved by using a more 'biological' and less 'statistical' language in some parts of the result section. For example, 'As predicted, anxious and calm individuals were as likely to solve the task' is clearer and easier to follow than 'In contrast, but in support of prediction 1a, anxiety was unrelated to task-solving ' (line 276). The first sentence of the result is also a good example of what I consider clearer (line 272). Similarly, the authors could clarify the results if they avoid referring to models by their numbers, but focus instead on describing the trends. For example 'In both the models for anxiety (a) and boldness (b) (models 3NF a and b), the three-way interaction was significant ' (line 292), is obscure compared to the following sentences describing the nature of these triple interactions. Also, the authors could avoid referring to the 'response' (line 293, 296 for example) and use the actual variable instead (see also this issue in the discussion).

Throughout the section, please present the relevant statistics in text for each result you presented. This will provide a better link between the text and the tables. Please provide full model components (deviance, size and significance of variance components, degrees of freedom, number of observations and groups) in tables. These are useful to show whether the models are adequate. For example, the generalized linear mixed models could be biased by over dispersion (i.e. is the residual deviance close to the number of observations?).

The age effects on social learning is indeed interesting. One could actually hypothesize that the younger individuals would rely more on social learning to assess the risk in their environment. Presenting these results more thoroughly through out the three subsections of the results would make this aspect of the study more integrated. It would also put more focus on these interesting results. In addition, a seemingly important aspect missing in the manuscript is the effect of dominance rank on the ability of individuals to interact with novel/hidden foods, to act as demonstrator, or to use/acquire social information. Subordinates may be at the periphery of the group, less gregarious for example, and thus unlikely to have the same opportunities than dominant individuals. Alternatively, dominant individuals can more easily acquire a food item after the task is solved (at least for the novel food experiment) and so may invest less resources into solving the task at first, compared to subordinate. How did dominance affect the patterns reported in this study? At present the authors 'corrected' for the confounding effects of this variable but the actual patterns are also interesting. I suggest either briefly discussing it in the discussion or integrating it in the objectives of the study and adequately presenting the results (the statistical tests, effect sizes in text, supported by a figure).

Line 36: I would move '(the demonstrators)' to the first point of the description (line 34). This would avoid a potential confusion where the individuals acquiring social information would be taken as the demonstrator.

Line 38 to 44: I am just being picky and I understand what the authors mean here. However, I feel this sentence uses unnecessary terminology. Why not simply write that an individual's boldness and exploration (or neophobia) affect its tendency to approach novel objects/food? Or even better why not not just write that an individual's response to novel stimuli may affect its ability to acquire personal information?

Line 53: 'personality' is a bit vague here. Perhaps the authors could be more direct and simply present the traits they measured?

Line 129: Typo, remove one of the ')'.

Line 185: what is IQR standing for?

Line 201 to 203: A seemingly important covariable would be also the rank order of the focal individual relative to the subgroup in which it is tested (the authors present this as a complication lines 159 to 168). We can expect subordinate individuals to be less likely to approach a novel/hidden food in the 'retest' portion of the trial compared to dominant ones?

Line 190 to 210: please present the dependent variables used in these models in text.

Line 275: What is 'convergent validity'? Please define, or use a simpler term.

Line 298 to 300 is a good example where presenting the formal statistics in parentheses would help make the point clearer.

Line 349 – 350: this is an important and interesting aspect of this study that is not put forward in the results. Perhaps presenting an additional figure illustrating the effect of time spent watching the demonstrators on the change in response toward the tasks would be a good addition to the manuscript. Also, why is the time spent spent having and inconsistent effect?

Line 354 – 355: Maybe the authors should nuance the tone to acknowledge that the third explanation is the most likely of the three, but that the authors do not have any conclusive results on this.

Line 368 : I guess sample size was limited to investigate the effect of the demonstrator's phenotype with the current dataset? It would be worth briefly mentioning why the authors did not investigate this aspect of the topic, given that the data is available.

Line 382: 'Specifically, while sex is unrelated to boldness in our population, age is not: juvenile baboons are bolder than adults (unpublished data)' This is the type of background information that is needed in the methods. The following statement also presents some important information that should be presented in the result section. What was the age, personality, dominance rank, etc of demonstrators vs the average of the group? Vs the focal individuals? Note also that the sentence implicitly uses a 'double negation'. Perhaps rephrase to a more direct sentence.

·

Basic reporting

Overall, I thought the basic reporting was fine. There were some aspects here that could be improved, which I will note below.

1. In the Introduction, I would not consign the hypotheses/predictions to a table, especially as a similar table is used to summarize the predictions and results in Table 3, which is fine, especially as its discussed a bit in the text already.

2. In describing the studies/experiments, it would be better, I think, to have each be reported as a separate methods section with subjects, manipulations, etc. An additional layer of subheadings would help, too. My reason for suggesting this is, as it stands, I found it a bit a hard to follow what was going on.

3. The English is mostly clear, though I have a few suggestions for improvement. First, there are a lot of instances throughout in which parentheses are used. In many of these cases, I would revise the sentence to avoid use of parentheses, even if it means splitting the sentence in some way. Also, the authors could work to tighten the prose a bit throughout.

4. On page 4 there were some blank boxes and I could not work out what they referred to.

5. The note about animal welfare (lines 265-270) does not belong in the section where it is placed, but probably in the methods section or its own section.

Experimental design

The experimental design was fine, especially given the limitations of conducting naturalistic experiments. Ethical standards have been met. From my reading, it appears as if the research questions could be addressed using the methods described. I am also fairly confident that the experiments adequately address the research questions, which were well-defined and meaningful.

Validity of the findings

Overall, the findings are robust and clearly laid out and the results are discussed in terms of how they relate to the research question. i don't think the authors go beyond their data

One recommendation in terms of the analyses. The authors categorized boldness and anxiety, namely to conform to methods used by others. I would strongly advise against doing so. It reduces power and when used in interaction terms, this practice can lead to spurious results. A good summary is Royston P, Altman DG, Sauerbrei, W. 2006. Dichotomizing continuous predictors in multiple regression: A bad idea. Statist Med 25:127-141.

While using continuous personality predictors will not make the results 'strictly comparable' to other in the sense of analyses being conducted, anybody who seeks to conduct a meta-analysis or estimate the size of your effect if using a mean split (especially if the personality scores are z-transformed first and ranges are provided) should have no problem in getting a comparable effect size. Finally, the author can note the problems associated with categorizing variables.

On page 10 (line 198) the authors refer to a p <= .05. I think it would be more appropriate to just stick to p < .05 as is convention, unless I am missing something.

Some additional thoughts. In the beginning of the Discussion the authors note that anxious individuals attended more to the social information than calm individuals. Might it be that this represents something like vigilance as opposed to curiosity? Might something else be related to the tendency for bolder individuals collecting social information? It would be good to present some follow-on hypotheses complete with ideas on how to test them.

---

## Round 0.2 · accepted · Accept

The authors are commended for the very thorough revision performed. The structure of the manuscript (related to predictions and hypotheses, and the results) and the details of the categorization of individuals are much clearer now. As a result, your manuscript is now acceptable for publication in Peer J.